# OpenReview forum: "PutnamBench: A Multilingual Competition-Mathematics Benchmark for Formal Theorem-Proving"
_ICML.cc/2024/Workshop/AI4MATH — ICML 2024 Workshop AI4MATH Oral_

### Official Review · Reviewer_1PtM · 2024-06-12

**Rating:** 6
**Confidence:** 3

**Summary:**

The paper introduces PUTNAMBENCH, a benchmark to evaluate neural theorem-provers using over 1100 formalized problems from the William Lowell Putnam Mathematical Competition. It supports Lean 4, Isabelle, and Coq, covering a wide range of undergraduate-level mathematics. PUTNAMBENCH aims to overcome limitations of existing benchmarks by providing diverse and challenging problems, separated solutions, and supporting multiple proof assistants, thus advancing automated theorem-proving research.

**Questions:**

1. What is the current SOTA LLMs' performances on the proposed dataset?

**Reasons To Accept:**

1. **Novel and Useful Dataset:** The proposed dataset provides a unique and comprehensive benchmark for evaluating neural theorem-provers, featuring over 1100 formalized problems from the prestigious William Lowell Putnam Mathematical Competition. It supports multiple proof assistants (Lean 4, Isabelle, and Coq), covering a broad range of undergraduate-level mathematics, addressing limitations of existing datasets.

2. **Clear and Easy-to-Follow Writing:** The paper is well-written, with a clear and concise presentation of the benchmark's design, implementation, and potential impact, making it accessible and valuable for researchers in the field of automated theorem proving.

**Reasons To Reject:**

Lack of Baseline Experiments: The biggest limitation of the paper is the absence of experimental results. There are no baseline experiments showcasing the performance of current LLMs or human performance on PUTNAMBENCH, which is crucial for understanding the benchmark's effectiveness and the current state of theorem-proving capabilities.

---

### Official Review · Reviewer_1y4M · 2024-06-12

**Rating:** 8
**Confidence:** 5

**Summary:**

This work introduces a multilingual benchmark for formal theorem proving. The benchmark, called PutnamBench, consists of 421 problems in Lean 4 and Isabelle, and 250 problems in Coq. All problems come from the William Lowell Putnam Mathematical Competition between 1969 and 2023. PutnamBench consists of over 1100 formal theorem proving tasks in total.

**Questions:**

Why no performance results are provided? Is it because of time constraint?

**Reasons To Accept:**

1. The paper is well motivated since more challenging benchmarks are necessary for increasingly performant LLMs.
2. The formalized problems are new in all 3 formal languages (Lean 4, Isabelle and Coq) and could be a valuable source for future benchmarking in this community
3. The writing is clear in general and very easy to follow
4. Comparison is given among previous similar benchmarks and it can be seen that PutnamBench has several advantages (although the columns Metamath and HOL Light are non-existant).
5. Related work is well presented in a logical manner

**Reasons To Reject:**

1. No baseline performance is provided. We don't know how the strongest baselines (e.g. LEGO-Prover, DeepSeekProver, Lyra, HTPS, etc) will perform on these problems.

---

### Official Review · Reviewer_no6W · 2024-06-12

**Rating:** 9
**Confidence:** 4

**Summary:**

PutnamBench presents a new data at the intersection of formal and informal theorem proving. Notably, the authors' dataset includes problems, of varying difficulty, annotated with proofs spanning a range of languages. The dataset promises to serve as a valuable benchmark and exploratory asset for the field, particularly, around "multilingual" mathematics.

**Questions:**

Could the authors please add more details on the students who provided the proofs? Are the authors sure the proofs are "quality"? Are the students maths-students or AI students exposed to formal maths?

Typo on lines 93-94?

**Reasons To Accept:**

PutnamBench is a creative and surely impactful new dataset. The multilinguality about the dataset in particular promises to advance the field and be a valuable resource for academic researchers. I'm already excited to explore the data myself!

It's clear the authors put substantial work into the data collection effort. I also appreciate the authors' clarity of writing -- the paper is very well-written.

**Reasons To Reject:**

The paper should not be rejected. As a minor point of feedback -- I felt that the paper is an interesting case where the related work section is perhaps a bit too long! The subsection on Methods for Formal Theorem Proving was a bit out of place here, as the authors just introduce the data, not actually evaluate said methods. However, that subsection will be useful in a broader piece that does evaluate (or propose new) methods to do well on PutnamBench.

---

### Meta-Review · Area_Chair_wMAP · 2024-06-13

**Recommendation:** Accept (Oral, top 2)
**Confidence:** 5

**Metareview:**

Great paper. It introduces a new dataset at the intersection of formal and informal theorem proving, which can serve as important and challenging benchmarks necessary for increasingly performant LLMs. The paper is well-written, and the creation of the dataset is well-motivated. The only drawback is the lack of baseline experiments on the newly proposed dataset.

---

### Decision · Program_Chairs · 2024-06-13

Accept (Oral)